# An Assessment of a New Rapid Multiplex PCR Assay for the Diagnosis of Meningoencephalitis

**DOI:** 10.3390/diagnostics14080802

**Published:** 2024-04-11

**Authors:** Genoveva Cuesta, Pedro Puerta-Alcalde, Andrea Vergara, Enric Roses, Jordi Bosch, Climent Casals-Pascual, Alex Soriano, Mª Ángeles Marcos, Sergi Sanz, Jordi Vila

**Affiliations:** 1Department of Clinical Microbiology, Hospital Clinic, 08036 Barcelona, Spain; gcuesta@clinic.cat (G.C.); vergara@clinic.cat (A.V.); jobosch@clinic.cat (J.B.); mmarcos@clinic.cat (M.Á.M.); 2Department of Infectious Diseases, Hospital Clínic-IDIBAPS, University of Barcelona, 08007 Barcelona, Spainasoriano@clinic.cat (A.S.); 3Department of Basic Clinical Practice, Faculty of Medicine, University of Barcelona, 08007 Barcelona, Spain; sergi.sanz@isglobal.org; 4CIBER Enfermedades Infecciosas (CIBERINFEC), Instituto de Salud Carlos III, 28029 Madrid, Spain; 5Institute of Global Health of Barcelona (ISGlobal), 08036 Barcelona, Spain; 6CIBER Epidemiología y Salud Pública (CIBERESP), Instituto de Salud Carlos III, 28029 Madrid, Spain

**Keywords:** meningitis, encephalitis, FilmArray ME, QIAstat-Dx ME, multiplex PCR

## Abstract

The rapid and broad microbiological diagnosis of meningoencephalitis (ME) has been possible thanks to the development of multiplex PCR tests applied to cerebrospinal fluid (CSF). We aimed to assess a new multiplex PCR panel (the QIAstat-Dx ME panel), which we compared to conventional diagnostic tools and the Biofire FilmArray ME Panel. The pathogens analyzed using both methods were *Escherichia coli* K1, *Haemophilus influenzae*, *Listeria monocytogenes*, *Neisseria meningitidis*, *Streptococcus agalactiae*, *Streptococcus pneumoniae*, Enterovirus, herpes simplex virus 1–2, human herpesvirus 6, human parechovirus, varicella zoster virus, and *Cryptococcus neoformans/gattii*. We used sensitivity, specificity, PPV, NPV, and kappa correlation index parameters to achieve our objective. Fifty CSF samples from patients with suspected ME were included. When conventional methods were used, 28 CSF samples (56%) were positive. The sensitivity and specificity for QIAstat-Dx/ME were 96.43% (CI95%, 79.8–99.8) and 95.24% (75.2–99.7), respectively, whereas the PPV and NPV were 96.43% (79.8–99.8) and 95.24% (75.1–99.7), respectively. The kappa value was 91.67%. Conclusions: A high correlation of the QIAstat-Dx ME panel with reference methods was shown. QIAstat-Dx ME is a rapid-PCR technique to be applied in patients with suspected ME with a high accuracy.

## 1. Introduction

Central nervous system (CNS) infections can be caused by bacteria, viruses, or fungi and are clinical entities associated with high morbidity and mortality [1,2,3]. Infections like these are frequently encountered in clinical and emergency department settings, presenting challenges due to their highly variable clinical manifestations, the significant differences in CSF characteristics between viral and bacterial infections, and the complexities involved in establishing a prompt and accurate diagnosis.

Therefore, to optimize directed therapy and hopefully improve patient outcomes, the early and accurate identification of the etiological agent is critical [3,4,5]. This has been made possible through the development of multiplex PCR (M-PCR) tests to detect the most common microorganisms causing encephalitis, meningitis, or meningoencephalitis (ME) in cerebrospinal fluid (CSF) [6]. At present, there are two M-PCR tests on the market: the Filmarray ME Panel, BioFire Diagnostics (Salt Lake City, UT, USA), launched in 2015, which is based on a nested PCR followed by a melt curve analysis in a microarray format, and the recently introduced QIAstat-Dx ME panel cassette (QIAGEN, Hilden, Germany), which is based on a multiplex real-time PCR platform. Both methods target potential ME pathogens in CSF, *Escherichia coli* K1, *Haemophilus influenzae*, *Listeria monocytogenes*, *Neisseria meningitidis*, *Streptococcus agalactiae*, *Streptococcus pneumoniae*, enterovirus, herpes simplex virus 1–2 (HSV-1 and 2), human herpes virus 6 (HHV-6), human parechovirus (HPV), varicella zoster virus (VZV), and *Cryptococcus neoformans/gattii*. The QIAstat-Dx ME panel has two additional bacterial targets which are for the detection of *Mycoplasma pneumoniae* and *Streptococcus pyogenes*, whereas the Filmarray ME panel has a target for CMV. The main purpose of this study was to assess the new QIAstat-Dx ME panel using comparisons with data obtained using a FilmArray ME panel, using conventional laboratory diagnostic methods and clinical diagnosis as a gold standard. 

## 2. Materials and Methods

### 2.1. Study Design and Clinical Samples

This study was conducted using 50 consecutive CSF samples from patients with a clinically suspected central nervous system infection which were analyzed in parallel using Gram staining, a Filmarray ME panel (BioFire Diagnostics; Salt Lake City, UT, USA, and conventional methods, meaning culture, antigen detection, and rt-PCR methods (see Section 2.2), as indicated in algorithm Figure 1. The remaining volume of each sample was frozen at −20 °C to be thawed and analyzed later using the QIAstat-Dx ME panel (QIAGEN, Hilden, Germany). Positive samples were frozen at −20 °C for a period lasting between 1 and 6 months The motivation for this analysis stemmed from the potential advantages of the QIAstat-Dx ME test, particularly its capability to perform amplification curve analyses and assess the cycle threshold (Ct) value upon obtaining a positive result. 

CSF cytology and biochemistry results, the final diagnosis tools of this study, were collected retrospectively. The initial decision to use the Filmarray ME panel was made by the attending physician after consulting with specialists in infectious diseases and/or microbiology, guided by a clinical suspicion of ME. 

### 2.2. Conventional Methods

The conventional microbiology protocol for CSF samples with a suspicion of ME includes the inoculation of blood and chocolate agar, thioglycolate broth, an *S. pneumoniae* antigen (BinaxNOW *S. pneumoniae* Antigen Card, BinaxNOW, Abbott, Chicago, IL, USA), and the detection of a *Cryptococcus neoformans* antigen (Remel™ *Cryptococcus* Antigen Test Kits, Thermo. Scientific, Lenexa, KS, USA). Complementary tests were those that were performed additionally on the CSF samples or those in which discrepancies were observed. They included 16S rRNA amplification and Sanger sequencing in CSF (SensiFAST™ SYBR Hi-ROX kit, Meridian Bioscience, Inc., Cincinnati, OH, USA), and sequences were identified using the Blast algorithm in the National Center for Biotechnology Information [NCBI] database, *Neisseria meningitidis* antigen detection (latex agglutination Wellcogen™ *N. meningitidis*, Thermofisher, Waltham, MA, USA), pathogen isolation in blood cultures (BACTEC™ FX; BD^®^, NYSE, New York, NY, USA), and the detection of the *S. pneumoniae* antigen in urine (BinaxNOW *S. pneumoniae* Antigen Card, BinaxNOW, Abbott, Chicago, IL, USA). Conventional viral detections were performed by real-time PCR for HSV1/2, CMV, HHV-6, and VZV (Nanogen Advanced Diagnostics, Palex^®^, Barcelona, Spain) and enterovirus (OneStep RT-PCR Kit, QIAGEN ^®^, Hilden, Germany). 

The analytical limits of detection for viruses using conventional methods were 119 copies/mL for herpes viruses, 69 copies/mL for VZV, 183 IU/mL for HHV-6, and 3.2 copies/mL for enteroviruses. 

### 2.3. Multiplex PCRs 

A volume of 200 µL of CSF was used for both the Filmarray ME, (BioFire Diagnostics Salt Lake City, UT, USA), and QIAstat-Dx ME (QIAGEN, Hilden, Germany) analyses. To keep the processes as similar as possible, all samples were handled in the same biosafety hood in the microbiology laboratory with the necessary safety and hygiene measures for handling this type of sample, cleaning the hood after processing each sample. Both techniques were performed according to the manufacturer’s instructions.

The M-PCR results were considered true (negative and positive) if they were consistent with the results obtained by conventional methods. 

The detection limits for viruses for the Filmarray ME and QIAstat-Dx ME methods were 281 and 250 TCID50/mL for HSV-1, 28 and 50 TCID50/mL for HSV-2, 170 copies/mL and 1660 copies/mL for VZV, and 31,300 copies/mL and 10,000 copies/mL for HHV-6, respectively. For the enterovirus, both had a limit of detection of 5 TCID50/mL. This information was obtained from the manufacturers.

### 2.4. Definitions and Final Diagnosis Assignment

ME was defined by the presence of an inflammatory process of the brain in association with clinical evidence of neurologic dysfunction and/or signs of meningeal irritation. The final diagnosis of an episode was made by the investigators (G.C., P.P-A., and J.V.) after a thorough evaluation of the microbiological and radiological results, clinical evolution, response to treatment, and the presence or absence of an alternative diagnosis.

### 2.5. Statistical Analysis

The sensitivity of the test was calculated as (true positive, TP)/(TP + false negative (FN)), and the specificity was calculated as (true negative (TN))/(TN + false positive (FP)). The positive likelihood ratio (pLR) was calculated as sensitivity/(1–specificity), and the negative likelihood ratio (nLR) was calculated as (1–sensitivity)/specificity. The positive predictive value (PPV) was calculated as TP/(TP + FP) and the negative predictive value (NPV) was calculated as TN/(TN + FN). Accuracy was calculated as (TN + TP(TP + FP + FN + TN) [7]. Cohen’s kappa coefficient test was used to assess the level of agreement between the different assessment methods, Filmarray ME and QIAstat-Dx ME, and the conventional methodology. Classification of kappa values included “poor” (0.00), “slight” (0 to 0.20), “fair agreement” (0.21 to 0.40), “moderate agreement” (0.41 to 0.60), “substantial” (0.61 to 0.80), and “complete agreement” (>0.8). Data were analyzed using Stata Statistical Software Release 18 (StataCorp, College Station, TX, USA). A p of less than 0.5 was considered a statistical significance level. The sensitivity, specificity, positive predictive value (PPV), negative predictive value (NPV), and kappa correlation index were calculated, considering the conventional methods as the gold standard. Both techniques yielded a result of “Not detected” or “Invalid” when the result was negative or invalid, respectively. FilmArray showed the positive results as “detected” and provided melting curves. QIA/ME showed positive results as “detected” and provided a cycle threshold (Ct).

## 3. Results

Fifty CSF samples were analyzed. Table 1 presents the results, including the CSF cytological and biochemical characteristics. Pathogens were detected by conventional methods in 29 samples (58%): 12/29 (41%) were bacterial (five *S. pneumoniae*, three *L. monocytogenes*, two *S. agalactiae*, one *N. meningitidis*, and one Acinetobacter baumannii), and 17/29 (59%) were viral cases of ME (seven HSV-1, four VZV, three HSV-2, two HHV-6, and one enterovirus). Twelve samples were determined to be negative by conventional methods. In 32 (64%) cases, there was concordance between all three methods: conventional methods and Filmarray ME and QIAstat-Dx ME panels.

The sensitivities and specificities were 96.5% (CI95%, 79.8–99.8) and 95.4% (CI95%, 75.2–99.7), respectively, for the QIAstat-Dx ME panel, with complete agreement with the conventional method (91.8%) according to Cohen’s kappa index and 85.19% (CI95%, 55.9–90.2), and 57.14% (CI95%, 29.6–70.3), respectively, for the Filmarray ME panel; according to Cohen’s kappa, we find a moderate agreement with the conventional method (43.5%) (Table 2). These statical parameters were calculated versus the gold standard and taken into consideration the final clinical diagnosis.

The Filmarray ME panel reported seven CSF samples with single-pathogen false positive results and five CSF samples with polymicrobial false positive results. Four were false negative results. The QIAstat-Dx ME panel reported only one false positive and one false negative result. The false positives reported by Filmarray ME were as follows: nine HSV-1, two *H. influenzae*, two *S. agalactiae*, two *S. pneumoniae*, one *E. coli* K1, one CMV, and one HSV-2. The false negatives included three HSV-1 and two *S. pneumoniae* results. The only false positive result reported by the QIAstat-Dx ME panel was a VZV result, while the only false negative was an HSV-1 result. Table 3 shows the discrepancies observed between the M-PCR diagnostic techniques and conventional methods.

## 4. Discussion

The early and accurate identification of the etiological agent causing ME is crucial for patient management [1,2]. A meta-analysis of 13 articles [8] showed high sensitivity (90%) and specificity (97%) for Filmarray ME results; however, there are still doubts about the reliability of certain M-PCR results in clinical practice [9,10,11]. Trujillo-Gomez et al. conducted another meta-analysis including 19 studies and found high specificity for the technique. However, the sensitivity varied from 89.5% to 93.5% depending on the reference method used [12]. 

Recently, Humisto et al. [13] published a paper comparing Filmarray ME and QIAstat-Dx ME techniques for the early diagnosis of ME, reporting that Filmarray ME was more reliable than QIAstat-Dx ME, with 0% and 6.5% error rates, respectively, and concluded that the BioFire FilmArray meningitis/encephalitis panel produced more positive results than the QIAstat-Dx meningitis/encephalitis panel in herpesvirus analyses. However, no clinical data were taken into consideration in Humisto’s study. Our study was in agreement with the above-mentioned results, showing a higher number of positive samples for herpesvirus in the Filmarray panel than in the QIAstat-Dx panel; however, we considered them false positives since our specific PCR results for herpesvirus were negative and, in addition, the final clinical diagnosis in all these samples but one was not herpetic encephalitis. In contrast, another recent publication showed comparable performance between both panels without significant differences [14].

Some previous studies have reported similar results concerning the use of the Filmarray ME panel. Johan Lindström et al. [9] analyzed 4199 CSF samples and obtained a sensitivity for HSV-1 of 82.4%. Amy L. Leber et al. [11] reported low sensitivities for some viruses, especially for HHV-6 (85.7%). False positives can be explained by two different reasons: (i) the CSF collection by lumbar puncture may have been traumatic and we are actually detecting traces of pathogenic genetic material present in the blood and not in the CSF [15], as may be the case for herpesviruses; however, if this scenario occurs, it should still be detected by conventional methods; and (ii) accidental contamination has occurred at some point in the process. In patients with suspected ME, findings of a false positive, especially for herpes simplex, are problematic because they may lead to unnecessary treatment with consequent drug toxicity [16]. In addition, false positive findings may complicate the search for other possible explanations for the clinical picture.

With the FilmArray ME panel, the most frequent false positive we obtained was for HSV-1, which has also been reported in other series [17]. In contrast, HSV-1 was the main pathogen involved in false negative results in other studies [8,9,18]. The microbiological ability to interpret a positive FilmArray ME result is very limited since we can only observe the melting curve, making it difficult to differentiate between true positives and false positives or contaminations [18]. It should be noted that the microorganisms responsible for most of the false positives obtained in our study coincide with those of other studies [9,11,19,20,21]. In a review by Trujillo-Gomez et al. [12] including 7090 CSF samples, the respective sensitivities and specificities were 87.5% and 98.5% for *S. pneumoniae*; 71.5% and 99.5% for *S. agalactiae*, and 75% and 99% for HSV-1. In addition to the better overall performance of the QIAstat-Dx ME panel in this study, one additional advantage of this method is that once a microorganism is detected, the amplification curve and Ct value can be assessed (Appendix A). This can be helpful in clinical interpretation. In our cohort, only one false positive was obtained by the QIAstat-Dx ME panel for VZV. In this case, the QIAstat-Dx ME panel showed a correct sigmoidal curve with a Ct of 38.5. The patient had already been diagnosed with VZV ME 15 days earlier in another hospital. This clinical picture, together with the high Ct value obtained, allowed this case to be interpreted as the detection of remnants of past ME. The patient was finally diagnosed with VZV-associated vasculitis, which responded properly to treatment with methylprednisolone, acetylsalicylic acid, and cyclophosphamide. Appendix A shows the difference between the Ct of this patient and that of a case of active VZV encephalitis. This case leads to the need to point out that despite the undoubtedly great usefulness of M-PCRs, the user must be trained in the interpretation of the results and must carefully place them in the context of the individual patient.

The false negative results obtained via M-PCR techniques can be explained for two different reasons: (i) the pathogen causing the ME is not included in the M-PCR targets, and (ii) the pathogen is included among the M-PCR targets but still not detected. ME caused by a pathogen not included as an M-PCR target is one of the main limitations of these techniques. In this sense, a negative M-PCR should not exclude an ME diagnosis in cases of high clinical suspicion, especially in patients with an increased probability of an “atypical” ME cause, such as immunosuppressed or neurosurgical patients. Accordingly, we detected a case of ME in which both M-PCR results were negative but *A. baumannii* was isolated by conventional methods. Most data regarding false negatives with the Filmarray ME panel report a low or suboptimal sensitivity for detecting *C. neoformans/gatii*, HSV-1/2, VZV, and enteroviruses [8,11,12]. In our study, the false negative results with the FilmArray ME panel were mainly with HSV-1 (three) and *S. pneumoniae* (one). With the QIAstat-Dx ME panel, we only obtained a single false negative (HSV-1).

Finally, as for invalid results, we obtained only one with a very purulent CSF sample on the FilmArray ME panel (*S. pneumoniae* detected by conventional methods and QIAstat-Dx ME). 

This study has the following limitations: Samples were analyzed at different times with intermediate freezing. Samples that showed false positive results obtained using the Filmarray ME panel were not reanalyzed using the same technique due to the volume limitation of the CSF samples, with a preference for using another technique to confirm the results. This same volume limitation meant that the samples tested using the QIAstat-Dx ME panel were selected on the basis of availability and, since we wanted to evaluate the performance of the QIAstat-Dx ME panel against false positives and false negatives, we included a wide range of positive CSF samples, which does not represent the real situation in a clinical laboratory, where most CSF samples are negative. In addition, there are targets that could not be studied due to a lack of positive CSF samples. Furthermore, the effectiveness of both the Filmarray ME and QIAstat panels may have been constrained due to frequent use in patients with normal or minimally altered CSF characteristics, where false positives could pose issues. Finally, while the benefits of utilizing a rapid, highly sensitive, and specific molecular test for diagnosing ME are undeniable, evaluating the clinical and economic impacts of such methods was beyond the scope of this study.

## 5. Conclusions

The use of M-PCR in microbiology laboratories for the early detection and treatment of ME does not exempt culture and subsequent analyses by conventional methods. M-PCR panels, although they have a high cost, offer important advantages, such as a faster turn-around time, which can impact the management of the patient. The QIAstat-Dx ME is easy to use and has a fast turnaround time, and it shows a higher sensitivity and specificity (96.43% and 95.24%, respectively) as well as positive-predictive and negative-predictive values (96.43% and 95.24%, respectively). In addition, the results obtained are accompanied by an amplification curve and its corresponding Ct value, which allows for a better microbiological interpretation together with other clinical data. Conversely, while the Filmarray ME demonstrated high sensitivity (85.1%), it also yielded some false positives, yielding a low PPV of 71.8%. The additional performance of routine tests may be beneficial, especially in cases in which the diagnosis remains uncertain.

## Figures and Tables

**Figure 1 diagnostics-14-00802-f001:**
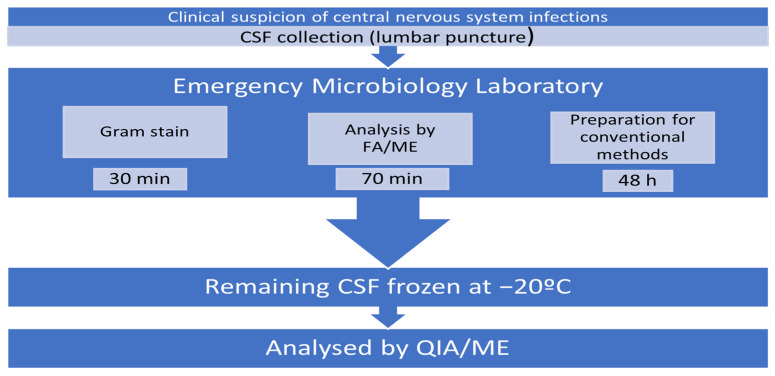
Sample processing algorithm.

**Table 1 diagnostics-14-00802-t001:** Summary of cases analyzed.

Sample	Age (Years)	Red Blood Cells (/mm^3^)	White Blood Cells (/mm^3^)	Neutrophils (%)	Lymphocytes (%)	Glucose (mg/dL)	Proteins (mg/L)	ADA (U/L)	LDH (U/L)	Conventional Methods	FilmArray ME	QIA-Stat-DX ME	Final Diagnosis
**1**	40	8	15	4	85	72	224	6.8	-	Enterovirus	Enterovirus	Enterovirus	Enterovirus encephalitis
**2**	69	100	930	32	65	66	252	27.3	-	*L. monocytogenes*	*L. monocytogenes*	*L. monocytogenes*	*L. monocytogenes* meningoencephalitis
**3**	48	8800	140	91	5	63	4844	21.4	107	*L. monocytogenes*	*L. monocytogenes*	*L. monocytogenes*	*L. monocytogenes* meningoencephalitis
**4**	60	130	22	45	53	60	560	-	37	*L. monocytogenes*	*L. monocytogenes*	*L. monocytogenes*	*L. monocytogenes* meningoencephalitis
**5**	68	100	1480	88	8	<1	733	-	-	*N. meningitidis*	*N. meningitidis*	*N. meningitidis*	Meningococcal meningitis
**6**	46	130	37	85	5	<4	6910	-	-	*S. agalactiae*	*S. agalactiae*	*S. agalactiae*	*S. agalactiae* meningoencephalitis
**7**	49	21,120	2195	94	6	33	5741	47.1	-	*S. agalactiae*	*S. agalactiae* + HSV-1 + HSV-2 + *H. influenzae* + *S. pneumoniae*	*S. agalactiae*	*S. agalactiae* meningoencephalitis
**8**	72	170	2065	95	4	<4	6000	48.9	391	*S. pneumoniae*	Negative	*S. pneumoniae*	Pneumococcal meningitis
**9**	64	280	12,960	98	0	203	8794	34.8	166	*S. pneumoniae*	Invalid	*S. pneumoniae*	Pneumococcal meningitis
**10**	65	20	560	92	7	15	7342	-	-	*S. pneumoniae*	*S. pneumoniae*	*S. pneumoniae*	Pneumococcal meningitis
**11**	50	0	3590	95	5	<4	8350	-	-	*S. pneumoniae*	*S. pneumoniae*	*S. pneumoniae*	Pneumococcal meningitis
**12**	38	40	35	30	70	9	7120	-	-	*S. pneumoniae*	*S. pneumoniae*	*S. pneumoniae*	Pneumococcal meningitis
**13**	50	40	0	0	0	78	826	7.4	24	HSV-1	Negative	HSV-1	Herpetic encephalitis
**14**	69	0	0	0	0	76	267	-	-	HSV-1	Negative	HSV-1	Herpetic encephalitis
**15**	44	0	20	-	-	88	834	7.1	-	HSV-1	Negative	Negative	Pulmonary source fever with associated low consciousness level in a low CD4 HIV patient.
**16**	74	0	20	1	20	50	70	-	-	HSV-1	HSV-1	HSV-1	Herpetic encephalitis
**17**	65	10	20	4	85	75	339	6.4	-	HSV-1	HSV-1	HSV-1	Herpetic encephalitis
**18**	45	10	40	-	-	62	391	7.5	-	HSV-1	HSV-1	HSV-1	Herpetic encephalitis
**19**	58	0	10	-	-	73	258	5.1	<20	HSV-1	HSV-1	HSV-1	Herpetic encephalitis
**20**	48	10	302	0	89	61	1294.9	8.9	-	HSV-2	*S. agalactiae* + HSV-2	HSV-2	Herpetic encephalitis
**21**	52	60	342	0	100	62	550	-	-	HSV-2	HSV-2	HSV-2	Herpetic encephalitis
**22**	51	5	80	6	85	38	2391	14.8	163	HSV-2	HSV-2	HSV-2	Herpetic encephalitis
**23**	35	130	155	0	100	57	548	5.2	<20	HSV-2	HSV-2	HSV-2	Herpetic encephalitis
**24**	59	0	5	-	-	58	597	9.9	27	HHV-6	HHV-6	HHV-6	Limbic encephalitis due to herpesvirus 6
**25**	75	1920	4	-	-	65	645	11.8	-	HHV-6	HHV-6	HHV-6	Herpesvirus 6 encephalitis
**26**	30	20	240	-	-	43	155	-	-	VZV	VZV	VZV	VZV encephalitis
**27**	62	10	10	0	100	70	782	6.8	-	VZV	VZV	VZV	VZV encephalitis
**28**	78	5120	778	1	88	58	4778	37.8	407	VZV	VZV + HSV-1	VZV	VZV encephalitis
**29**	77	30	22	0	100	111	674	34	11	Negative	Negative	VZV	VZV encephalitis
**30**	51	220	33	5	93	63	783	7.9	27	Negative	*H. influenzae* + *S. pneumoniae* + CMV	Negative	Post-vaccine myelitis
**31**	83	112	10	-	-	106	271	4.4	-	Negative	*S. agalactiae* + *E. coli* K1	Negative	Confusional syndrome
**32**	33	220	0	0	0	73	275	5.2	-	Negative	HSV-1	Negative	Cytokine release syndrome in a CAR-T recipient
**33**	63	30	160	0	0	260	49	58	339	Negative	HSV-1	Negative	Wernicke’s encephalopathy
**34**	64	100	0	0	0	79	441	10.9	-	Negative	HSV-1	Negative	Stroke
**35**	70	75	8	-	-	230	1080	10.8	43	Negative	HSV-1	Negative	Herpetic encephalitis
**36**	88	0	0	0	0	76	309	8.1	-	Negative	HSV-1	Negative	Hypoglycemic crisis
**37**	33	50	0	0	0	87	321	6.5	<20	Negative	HSV-1	Negative	Drug intoxication
**38**	56	0	0	0	0	61	632	7.7	<20	Negative	HSV-1	Negative	Brain metastases
**39**	30	190	0	0	0	79	631	24.9	188	Negative	Negative	Negative	MELAS syndrome
**40**	60	0	0	0	0	63	667	3	-	Negative	Negative	Negative	Post-COVID-19 encephalitis
**41**	20	1200	0	0	0	36	87	-	-	Negative	Negative	Negative	Confusional syndrome due to fever
**42**	27	17,280	0	0	0	55	353	-	-	Negative	Negative	Negative	Chronic meningococcemia
**43**	65	1980	0	0	0	38	908	11.2	85	Negative	Negative	Negative	Prior diagnosis of listerial meningoencephalitis; currently undergoing treatment
**44**	83	480	0	0	0	123	1800	-	-	Negative	Negative	Negative	Confusional syndrome
**45**	67	300	40	47	46	35	3190	32.2	271	Negative	Negative	Negative	Aseptic meningitis
**46**	53	0	0	-	-	92	242	-	21	Negative	Negative	Negative	Epileptic syndrome
**47**	64	1380	8	-	-	108	1303	-	-	Negative	Negative	Negative	Lymphoproliferative disease
**48**	52	630	437	85	10	123	516	-	-	*A. baumannii*	Negative	Negative	Postoperative meningitis due to *A. baumannii*
**49**	68	20	37	-	-	57	576	13.6	18.4	Negative	Negative	Negative	Autoimmune meningoencephalitis by anti-Mglur5
**50**	82	1440	5	74	19	107	639	-	-	Negative	Negative	Negative	Stroke

Abbreviations. HSV = herpes simplex virus; HHV-6 = human herpesvirus 6; VZV = varicella zoster virus; CMV = cytomegalovirus; CAR-T = chimeric antigen receptor T-cell; MELAS = mitochondrial encephalopathy, lactic acidosis, and stroke-like episodes. Missing results have been denoted with hyphens. In some cases, with low white blood cells, neutrophil and lymphocyte percentages were not determined. Pleocytosis (>10 leukocytes/mm^3^) and high protein CSF levels (>600 mg/dL) are commonly found in encephalitis and meningitis. However, normal CSF cells and protein levels can be found, particularly in early viral cases.

**Table 2 diagnostics-14-00802-t002:** Comparison of sensitivity, specificity, PPV, NPV, and kappa correlation index for both M-PCR diagnostic techniques compared to the conventional methods.

	QIAstat-Dx-ME	FilmArray-ME
**Sensitivity (%)**	96.5% (CI95%, 79.8–99.8)	85.1% (CI95%, 55.9–90.2)
**Specificity (%)**	95.2% (CI95%, 75.2–99.7)	57.1% (CI95%, 29.6–70.3)
**PPV**	96.4% (CI95%, 79.8–99.8)	71.8% (CI95%, 43.7–78.3)
**NPV**	95.2% (CI95%, 75.1–99.7)	75% (CI95%, 55.9–0.2)
**Kappa correlation index**	91.67% (*p* < 0.001)	43.48% (*p*: 0.001)

**Table 3 diagnostics-14-00802-t003:** Discrepancies observed between M-PCR diagnostic techniques and conventional methods.

	Positive Specimens (N)		
Pathogen	FilmArray ME	QIA-Stat Dx	Conventional Methods
*E. coli* K1	1 ^a^	0	CSF culture (−) CSF 16S rRNA PCR sequencing (−)
*H. influenzae*	1 ^b^	0	CSF culture (−) CSF 16S rRNA PCR sequencing (−)
	1 ^c^	0	CSF culture (−) CSF 16S rRNA PCR sequencing (−)
*S. agalactiae*	1	0	CSF culture (−) CSF 16S rRNA PCR sequencing (−)
	1	0	CSF culture (−) CSF 16S rRNA PCR sequencing (−)
*S. pneumoniae*	0	1	CSF culture (−) CSF Ag *S. pneumoniae* (invalid) CSF 16S rRNA PCR sequencing (+) Blood culture (+)
	0	1	CSF culture (+) CSF Ag *S. pneumoniae* (+) Blood culture (+)
	1 ^b^	0	CFS culture (−) CSF *S. pneumoniae* Ag (−) CSF 16S rRNA PCR sequencing (−) Blood culture (−) Urine *S. pneumoniae* Ag (−)
	1 ^c^	0	CFS culture (−) CSF *S. pneumoniae* Ag (−) CSF 16S rRNA PCR sequencing (−) Blood culture (−) Urine *S. pneumoniae* Ag (−)
CMV	1 ^c^	0	CSF CMV-PCR (−) Blood CMV-PCR (−)
HSV-1	0	1	CSF HSV-1 PCR (+)
	0	1	CSF HSV-1 PCR (+)
	0	0	CSF HSV-1 PCR (+)
	9 ^b,e^	0	CSF HSV-1 PCR (9−)
HSV-2	1 ^b^	0	CSF HSV-2 PCR (−)
	1 ^d^	0	CSF HSV-2 PCR (−)
VZV	0	1	CSF VZV PCR (−)

CSF: cerebrospinal fluid; RT-PCR: reverse transcription PCR; ME: meningoencephalitis. ^a^ Positive results FilmArray for *E. coli* K1 and *S. agalactiae* in the same sample. ^b^ Positive FilmArray results for *H. influenzae*, *S. agalactiae*, HSV-1, HSV-2, and *S. pneumoniae* in the same sample. ^c^ Positive FilmArray results for *H. influenzae*, *S. pneumoniae* and Cytomegalovirus in the same sample. ^d^ Positive FilmArray results for *S. agalactiae* and HSV-2 in the same sample. ^e^ Positive FilmArray results for VZV and HSV-1 in the same sample.

## Data Availability

The data presented in this study are available on reasonable request from the corresponding author due to privacy reasons.

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
