# Peer review of "An Assessment of a New Rapid Multiplex PCR Assay for the Diagnosis of Meningoencephalitis"

_diagnostics, 2024, doi:10.3390/diagnostics14080802_

Round 1

Reviewer 1 Report

Comments and Suggestions for Authors

Please see attached word document. 

Comments on the Quality of English Language

Overall reasonable but please check over some wording and sentence structure. 

Author Response

Reviewer #1

Cuesta et al.

Assessment of a new multiplex PCR assays for rapid diagnosis of meningoencephalitis

Peer review:

This is an important area of investigation in clinical microbiology, with an increasing move to molecular testing and syndromic testing being implemented with significant potential advantages. Cuesta et al, present their comparison of a new multiplex PCR for CSF testing – called “QIA-ME” and compare this to their conventional methods. They also compare it to “FA-ME”, however the context of this is not elucidated well. Overall- it is reasonably well written, Although this may not be the focus of the paper with low numbers of samples tested (compared to some other studies), this would be useful information in the context of your particular study. How for example does this study add to the data from Humisto et al?

The main difference between Humisto’s paper and our is that in our study we took into consideration the clinical data to finally evaluate the molecular diagnostics obtained with the respective panels used.

 Major / general comments:

- How was ME defined? Patient population and selection of samples is poorly described. While several tests can be performed on the same sample for a comparison- the sample collection needs methods need to be clearly stated and possible limitations discussed as it appears that a “convenience” sample of 50 CSFs was taken rather than true cases of suspected infective ME.

The definition of meningoencephalitis (ME) has been added to the manuscript (please see page 3, lines 121-125). The samples were not selected based on “convenience” as the reviewer suggested, otherwise there were consecutive samples in which the remaining sample was enough to perform the QIA-ME panel, since sometimes the volume of CSF was not enough to carry out the conventional methods, FA-ME panel and QIA-ME panel. The initial decision to conduct the Filmarray ME panel was made by the attending physician after consulting with specialists in Infectious Diseases and/or Microbiology, guided by the clinical suspicion of ME. We have added such information to the manuscript (please see page 2, lines 74-76). 

- This study does not really take into account in details of how the new diagnostic system may benefit the lab and clinicians at your institution, apart from just stating the testing parameters compared to the “gold standard”.

Overall, the implementation of a highly sensitive, specific and rapid molecular test contributes without doubt to better management of the patient mainly concerning the type of treatment to be administered. However, evaluating the clinical and economic impacts of such methods was beyond the scope of this study. We have added this as a potential limitation of this study (please see page 9, lines 266-271).

-Rationale for comparing the QIA-ME was not well discussed. The methods were incomplete, especially when it came to the introducing the new system in the lab- what training was done for example. The decision of the “final diagnosis” and interpretation of testing- which is not always straightforward – is not clear at all.

The motivation for this analysis stemmed from the potential advantages of the QIAstat system, particularly its capability to perform amplification curve analysis and assess the cycle threshold (Ct) value upon obtaining a positive result. We have added this information to the manuscript (please see page 2, lines 69-72). In our lab we already have implemented the QIA-Stat GI to detect gastrointestinal, therefore our technicians were already well trained in managing the cartridges to be used in the test, but in this case with CSF samples instead of stool samples.

We agree with the reviewer that stablishing a final diagnosis is often challenging in these patients. In this study, final diagnosis of the episode was determined by the investigators (G.C., P.P-A., and J.V.) after a thorough evaluation of microbiological and radiological results, clinical evolution, response to treatment, and the presence or absence of an alternative diagnosis. For example, if the patient had a positive result for HSV-1, but a normal Magnetic Resonance imaging and improved without specific treatment for herpes, was considered a false positive. We have added such information to the manuscript (please see page 3, lines 121-125).

-Inconsistent use of terms such as referring to the types of M-PCR in different ways, which makes the manuscript harder to read and a bit confusing.

This aspect has been reviewed and corrected.

-The main table showing all results of testing is full of errors and shows testing was done on patients that may not have had a suspicion of an infective etiology to account for their symptoms.

We apologize for these mistakes. The table has been revisited and the mistakes found have been corrected. Should the reviewer or editor identify any other potential errors, we would be happy to address and correct them accordingly.

Regarding the comment on testing patients with low suspicion of infective meningoencephalitis, we agree that this is a potential limitation. As already explained, the initial decision to conduct the Filmarray ME panel was made by the attending physician after consulting with specialists in Infectious Diseases and/or Microbiology. Despite this, many of the analyzed CSFs had normal glucose and protein levels and lack CSF pleocytosis. In this setting, false positive results might be problematic. We have added this issue as a potential limitation (please see page 9, lines 266-271).  

Limitations of this study and how to address these issues were not discussed in enough depth. This study really shows some testing parameters on a heterogenous population compared to “conventional testing” which includes culture, PCR and antigen testing. To really compare testing modalities between different testing assays and control as many parameters as possible it is often useful to “spike” (often with a dilution series) samples with a known pathogen and then test with all testing modalities being compared. This clearly was not done but there is probably information available from the companies selling these commercial assays.

The limitations have already been stated in the manuscript (lines 257-265). Concerning the spike of the samples with different microorganisms was not the objective of our study and in addition, as the reviewer said, this has probably been performed by the company before launching the assay.

A discussion of your findings to company data may be informative, although there is some good discussion of other studies looking at these tests commercial assays.

We do not understand exactly what the reviewer means in this statement.

Specific comments:

Line 16-17 “our conventional diagnostic tools”- this is vague, what do you mean by this? Culture? I understand this is elaborated in the text, but it does need to be briefly described here.

The conventional diagnostics tools have deeply been described in the manuscript (lines 86-105). We do not know why we should mention them here.

Line 20-21 “The statistical parameters were calculated”, this is a pointless statement if you then go on to describe the various parameters. It is often better to say something along the aims of “we aimed to compare the sensitivity, specificity (and other possible parameters) of X test to our conventional method of culture and compared to test Y”

Thanks, this has been changed accordingly. (page, 1; lines 24-25)

Line 23-35, please keep decimal points consistent. You only need to state the CI95% in the first brackets then the rest can be inferred.

This has been modified accordingly, (page, 1; lines 29-31)

Line 32-33 “Therefore, for correct treatment and patient outcome” I would rephase along the lines of: Therefore, to help optimise directed therapy and hopefully improve patient outcomes…

Following the reviewer’s suggestion, we have rephrased this sentence (please see page 1, lines 43-44).

Is there any evidence that these rapid diagnostics actually improve patient outcome, often patients with ME are broadly treated empirically for several suspect pathogens. Would testing help decrease drug toxicity or have other benefits?

There are some studies correlating the use of new multiplex real-time PCRs in CSF with reduced antibiotic treatment and toxicity (doi: 10.1007/s15010-018-1212-7). This seems logical, as, for instance at our institution, routine viral PCRs are performed only on weekdays. Consequently, a suspected case of herpetic encephalitis identified late on a Friday might not yield results until Monday. However, this is out of the scope of the present study. Should the reviewer or editor find it necessary we could add a comment or reference regarding this concept.

Line 38 “two rounds of nested PCR” Is not nested PCR two rounds by definition? I would just say based on nested PCR.

I agree with the reviewer, it has been changed.

Line 40: Yes, they both target these but maybe point out that the QIA-ME has two additional bacterial targets (total of 8) with Mycoplasma pneumoniae and Streptococcus pyogenes. The FA-ME has the addition of CMV for the viral pathogens -giving a total of 7 targets (vs 6).

These differences have been added into the manuscript.

Line 47: Is clinical diagnosis really a “gold standard”, how was this determined to limit any bias. Ideally you would want to determine testing parameters on an acute patient with undefined ME to your usual diagnostic processes. It is not clear why you compare the QIA-ME (apart from stating it is new). Context on whether other syndromic M-PCR are used in your institution and if the FA-ME is used in any context would be useful.

As already exposed, the final diagnosis of the episode was determined by the investigators after a thorough evaluation of microbiological and radiological results, clinical evolution, response to treatment, and the presence or absence of an alternative diagnosis. We think this reflects the common clinical practice.

I apologize, but I am not sure to understand what the reviewer means with the second part of the question. We had been performing Filmarray until we performed this study. Since then, we started using QIA-Stat.

Line 50-55: How were these samples chosen? How did you limit potential selection bias, where these consecutive samples from the emergency department or ICU etc. Please describe in greater detail.

How the samples were collected has been mentioned and added into the manuscript. As it has been previously mentioned these were consecutive samples with the exception of the samples from which we did not had leftovers due to the original small volume which obviously were not processed.

Again, please define your usual “conventional methods”. I see this is described in section 2.2 rather comprehensively but maybe summarise earlier in the article what you mean by “conventional methods” as this can differ lab to lab and also many would presume you mean, microscopy/ gram stain, and culture, but you are also including rapid antigen tests and real time PCR (which may not be seen as conventional by some).

The methods understood as conventional have been described according to the reviewer's comment in the manuscript.

It is uncertain why the remaining samples were frozen and thawed for the QIA-ME and not the FA-ME? Can this be discussed.

Because as we mentioned in the algorithm we use as a first approach for the microbiological diagnosis of a meningitis and encephalitis Gram-stain, Filmarray as a rapid test and conventional methods.

How was the “final diagnosis of each episode determined? – I presume a review of clinical records and microbiology results with a panel of experts? Who decided on this in uncertain cases? Sometimes detection of an organism has uncertain significance, especially in the clinical syndrome of infection ME.

As already mentioned, the final diagnosis of the episode was determined by the investigators Genoveva Cuesta, Pedro Puerta-Alcalde, and Jordi Vila) after a thorough evaluation of microbiological and radiological results, clinical evolution, response to treatment, and the presence or absence of an alternative diagnosis (please see page 3, lines 120-125). We agree that sometimes the detection of an organism has uncertain significance. That’s why we believe this approach based on clinical evolution, radiological results, and response to treatment, together with the microbiological results, might be more reliable.

Line 75-77 “The analytical limit of detection for viruses using conventional methods was 119 copies/mL for herpes viruses, 69 copies/mL for VZV, 183 IU/mL for HHV-6 and 3.2 copies/mL for enteroviruses.” Can you reference this? Also- what do you mean by herpes viruses, given VZV and HHV are also herpes viruses. 

This information was obtained from manufacturers’ instructions. This has been added into the manuscript. (Page 3, line 118-119).

Line 79-85: I’m not clear on if these tests were done at the same time or if the QIA-ME was done later, given it was from frozen samples?

As mentioned in the manuscript the samples were frozen and the QIA-ME panel applied later.

Line 86-89: You need to reference any data like this.

Again, this information was obtained from manufacturers.

Line 97: “between different imaging methods and intraoperative findings” – This sounds like it is coming from another study? Please revise.

There was indeed a mistake and the sentence has been changed.

Line 104: “I’m not sure if “gold standard” is the right term here, as this usually refers to the diagnostic method with the best accuracy. However, I understand that you mean it is the current “standard of care” and the denominator to determine whether new testing methodologies will advance our diagnostic accuracy. Again- as you also include standard real time PCR as your “conventional methods” this also confuses the picture. Often convention methods have been culture for bacteria but PCR for viruses given how they are hard to culture. I would think it is reasonable to compare the new tests to your current diagnostic methods, while acknowledging some of the shortfalls of culture and the more “conventional” diagnostic techniques including slower turn around PCR tests. I think making a comparison often should also include a standard of inoculum- such as spiked samples- which are then testing by conventional and novel methods.

I agree with the rational performed by the reviewer but we think that indeed by our conventional methods include all methods (culture, antigen and PCR) used to detect both viruses and bacteria.

Line 107: Most people would know Ct means “cycle threshold” but just state it in full.

Done

Line 111: Delete “ME”

Done

Line 111-112: Bacterial species in Italics please.

Done

Line 118-123: Please be consistent with your abbreviations: at times you refer to QIAstat-Dx ME panel then to the QIA- ME!

Thanks, it has been corrected.

Line 120-121: Again, conventional diagnostics may be less sensitive then M-PCR assays, although false positives can be an issue with M-PCR assays. “Taken into consideration the final clinical diagnosis”- how, who decided this? Some of the viral pathogens could be detected at times in the CSF but may not be the cause of active disease or may be concurrently detected but not the main pathogen.

This has already been answered (please see page 3, lines 121-125).

Line 136: “However, the” Just ends, ? reformat  

We do not understand what does it mean.

Line 185: I get concerned about comments like this as it defeats the purpose of the test which is for early diagnosis of the aetiology of ME, to help target therapy.

We agree that the early diagnosis of ME will help in a more adequate therapy, however we do not think that the comment misrepresents the purpose of the test.

Page 147-206: The discussion focuses a lot on the lab aspects- such as some pros and cons of multiplex syndromic testing. There needs to be some form of discussion about if introducing this test would allow faster turnaround time, reduced costs (they are not cheap) and result in better patient outcomes (something which is not clear at the time of writing). Also, there could be a discussion of how to manage discordant results between conventional and M-PCR methods.

These aspects mentioned by the reviewer has been incorporated into the discussion section of the manuscript. (Page 9, lines 274-276).

Page 209-217. I think the limited number of CSF samples in a rather heterogenous group of poorly defined “ME” patients – some where it looks like infection was not high on the differential, is a major limitation. Also- when doing a comparison like this it should ideally be done at the same time of in the clinical course – i.e. in the acute setting. I know that all tests were done on the same sample but it’s concerning that some testing was done after patients had been transferred from another hospital with a diagnosis already, the testing really should be done a useful clinical timepoint

What the reviewer described is the ideal scenario, but we think that the way that we perform the study is enough to show the accuracy of the new test. Also, all patients were treated and diagnosed in our hospital independently of later transfers to other settings. Finally, we believe there was a true clinical suspicion of meningoencephalitis, independently of the CSF characteristics. However, we agree that the only mildly altered CSF characteristics is a limitation of the study and have included it in the “limitations” section.

218-225: I think while you might have confirmed reasonable test performance parameters compared to usual CSF testing (“the standard care”), which has notable limitations- as it is you have not made a convincing case for introduction of either M-PCR assays and it is not clear the context of why you have compared the QIA-ME now? Has your institution been using the FA-ME in any capacity? Does your intuition use syndromic multi-plex testing in any other areas- such as on respiratory samples? If so, has this been a success or useful in your institution and is it cost effective and improving patient outcomes or anti-microbial stewardship?

The rational for comparing both multiplex PCR panels with our conventional methods has been mentioned before and added into the manuscript (page x, lines xxx-sss). We use the FLOW system from Roche to detect enteropathogens for patients with GI and also for community acquired pneumonia (This includes viruses plus atypical bacteria). In addition, we use QIA-Stat-Dx GI for screening of enteropathogens in stool donors since we have set up a stool bank in our hospital which provides stool samples to clinicians in our hospital or others who want to perform a fecal microbiota transplantation. Thee Flow system is cost-effective since we did an analysis previous to the introduction in the lab.

It would be nice to have a discussion of “where to” from here. With this retrospective study showing good testing parameters do you intend to follow up with a prospective clinical trial?

Based on the results presented in this study we have decided to change our routine workflow to diagnose meningitis/encephalitis and incorporate QIA-Stat-Dx ME instead of Filmarray ME.

Table 2:

Data missing from sample 10 and 11.

We apologize as this data was inadvertently lost. We appreciate the reviewer highlighting this oversight.

There is no clear data on the age of patients

Following the reviewer’s suggestion, we have added patients’ age in Table 2.

Many 0 values, when labs typically report <5 white blood cells. It is not clear what the “0” means in some parts of the table as they clearly should have differentiated the white cell count with high numbers of total cells.

In some cases, with low white blood cells, neutrophils and lymphocytes percentage was not determined. We substituted the zero with hyphens for clarity.

Some “?” which is unclear what this means.

This was a mistake. We have corrected it.

If no value available better to use a “-“ rather then just leave blank – as this looks like an omission.

We have done this, and clarified this point as a footnote.

Final diagnosis section is not consistent. “herpetic encephalitis” “HSV2 lymphocytic meningitis”

We apologize for this mistake. We have rewritten the diagnosis for better consistency.

Many unusual symptoms in the “final diagnosis section”- many are non-infective. I have concern that the selection of samples did not come from patients with a suspect infectious aetiology. Your definition of ME in patients is not defined and in the setting of no increased WCC is questionable. Detection of HSV with no elevated WCC may just be low grade reactivation and not meningoencephalitis.

As already exposed, we believe this reflects the clinical practice of a university hospital, where many patients experiencing fever and altered mental status, and then undergoing a lumbar puncture. In this setting, although CSF is only mildly altered, microbiological testing in commonly performed. Also, there exist some cases of clear herpetic encephalitis with normal CSF characteristics, especially when the time since initiation of symptoms is short.

Sample 15 – was this ME clinically?

Initially the patient had fever and was encephalitic. In view of the evolution and the negative result HSV-1 PCR, acyclovir was stopped.  

Sample 33- description is too long

We apologize. We have shortened it.

Sample 43- from your description I think this patient should be excluded! The CSF was not sampled at the time of presentation if this is an intra-hospital transfer.

The patient was transferred from another hospital. Lumbar puncture was performed due to worsened clinical status.

Sample 45- ? decapitated meningitis

This definition has been changed.

Reviewer 2 Report

Comments and Suggestions for Authors

Dear Editor,

I read with great interest the article titled 'Assessment of a new multiplex PCR assays for rapid diagnosis of meningoencephalitis,' written by Genoveva Cuesta and colleagues.

The article describes the analytical performance of a new commercial rapid multiplex PCR system (QIAstat-Dx ME panel) for the detection of infectious agents causing central nervous system infections in clinical samples of cerebrospinal fluid.

In the study, the comparative methods for evaluating the sensitivity and specificity of the QIAstat-Dx ME panel (test method) were clinical conclusions supported by the results of conventional methods, which included cerebrospinal fluid culture, antigen detection, and qPCR for specific viruses. Sanger sequencing of 16S rRNA was used as complementary testing. Samples were also submitted to a second, already established comparative rapid multiplex assay (FilmArray ME panel), which detects the majority of the same pathogens.

Considering the final clinical diagnosis and conventional methods as the gold-standard results, they found that the sensitivities and specificities were 96.5% (CI 95%, 79.8-99.8) and 95.4% (CI 95%, 75.2-99.7), respectively, for the QIAstat-Dx ME panel, and 85.19% (CI 95%, 55.9-90.2) and 57.14% (CI 95%, 29.6-70.3), respectively, for the FilmArray ME panel.

The observed results suggest that the test method not only exhibits good performance in the tested samples but also demonstrates a potential superiority over the comparative rapid multiplex PCR. The good adherence to the final clinical diagnosis also support its higher accuracy.

Given the critical nature of early and accurate identification of the etiological agent causing meningoencephalitis for patient management, documenting the clinical performance of this new assay at different sites is of global importance and great relevance.

Therefore, I recommend the acceptance of this manuscript for publication in the MDPI journal Diagnostics, contingent upon the authors making some adjustments to enhance its scientific suitability for the journal's readership.

These adjustments include ensuring terminology consistency throughout the manuscript, refining phrases for precision and clarity, addressing any potential gaps in the text, including the lack of some results, improving the organization of the text for better flow, explaining some presented results, and enhancing sentence clarity to ensure the message is conveyed effectively. I believe these modifications will significantly improve the manuscript's overall quality and readability.

I will provide the parts of the text that must be revised for these adjustments, following the organization of the text (IMRD), and classify them as major and minor changes. I will also offer suggestions to facilitate a clear and structured revision process, with the understanding that these suggestions are optional for the authors.

Title (suggestion): I suggest moving the word 'rapid' in the title to clarify the nature of the test evaluated in the study, emphasizing its 'bench to bed' approach. For example, 'Assessment of a New Rapid Multiplex PCR Assay for the Diagnosis of Meningoencephalitis. Additionally, if the authors wish to move from a more general title to one that includes the kit name and the specificities of the study, it could be more informative and explanatory.

Abstract (minor revision): Line 20 - 'The statistical parameters were calculated.' This statement appears somewhat isolated and lacks detailed information. Could the authors please clarify the specific statistical parameters calculated and how they were applied in the study? If this information is not critical to the understanding of the results, it may be more concise to remove this sentence.

Abstract (suggestion): I also recommend considering a revision of the abstract to reflect the suggested changes in this review.

Introduction: This section could be expanded slightly, as it currently consists of only one paragraph. Although the study's objective is clearly defined, more details could be included to provide readers with a broader background on the topic. For example, elaborating on the approach to a patient with central nervous system (CNS) infections would help to contextualize the study further. This could include the importance of the biochemical tests presented, as well as the different detection methods used in conventional methods.

Material and methods (major revision): In Line 53, it is mentioned that 'The remaining samples were frozen at -20°C, then thawed and analyzed by QIAstat-Dx ME panel.' To provide more clarity to the reader regarding storage conditions, it would be helpful to include the range of time the samples were frozen before being tested by the QIAstat-Dx ME panel, if this data is available. This information could clarify storage questions for the reader, even if this storage did not affect the results.

Material and methods (major revision): In Line 54, it is stated that 'CSF cytology and biochemistry results, the final diagnosis of the episode, were collected retrospectively.' It would be beneficial for the reader if the authors could clarify which specific cytology and biochemistry parameters were collected, where they are presented in the study, and why they are important for the study. This additional detail will enhance the understanding of the study's methodology and its relevance to the final diagnosis.

Material and methods (minor revision): Line 56 - Table 1. Sample processing algorithm. Table 1, titled 'Sample Processing Algorithm,' appears more like a figure than a traditional table. To enhance clarity, I kindly suggest renaming 'Table 1' to 'Figure 1' or an appropriate figure designation.

Material and methods (major revision): In the 'Conventional Methods' and 'Multiplex PCR' sections, the assays' limits of detection are described in different units, which could lead to confusion. For clarification, it would be beneficial to harmonize the units of measurement. For example, in Line 75, the analytical limit of detection for viruses using conventional methods is expressed in copies/mL for herpes viruses, VZV, and enteroviruses, and in IU/mL for HHV-6. In contrast, Line 86 describes the detection limits for viruses in FilmArray ME and QIAstat-Dx ME in TCID50/mL for HSV-1, HSV-2, and enterovirus, and in copies/mL for VZV and HHV-6. It would be helpful to use consistent units across both sections to enhance the reader's understanding of the assay sensitivities.

Material and methods (minor revision): While the manufacturers of the compared kits are mentioned in the abstract, it would be more appropriate and informative to also cite them in the '2.3. Multiplex PCRs' section. This section details the use of both the FilmArray ME and QIAstat-Dx ME analyses but does not specify the manufacturers' names, which are crucial for replicability and clarity.

Material and methods, and Results (major revision): The '2.4. Statistical analysis' section of the article is well-described and informative, explaining to the reader how all calculations were performed. Although these calculations are traditional and well-known, the detailed description is valuable as it saves the reader from having to consult other references for this information. However, the results of the Cohen's kappa analysis are mentioned only in the abstract and table 4, not in the results section. It would be worthwhile to include the outcomes of the kappa analysis in the text of the results section. This addition would enhance the completeness of the results and provide a clearer understanding of the level of agreement between the different methods used in the study.

Results (major revision): Table 2 – In the 'Results' section, particularly for Table 2, it would be beneficial for the authors to clarify, in the 'Materials and Methods' section, in the results, or in the legend of Table 2, the importance of each analyte measured. The authors should consider whether the biochemical parameters need to be included in this table or could be moved to a supplementary table. Regardless of the authors' decision, details about the importance of these measurements for the final diagnosis can be clarified at the same level of detail observed for the explanation of the statistical analyses. This would clarify their significance for the reader, avoiding the need to consult multiple references to understand their role in the final outcome. Since the final diagnosis is critical for comparison, these details deserve mention.

Results (major revision): line 117 and table 3 - In the second paragraph, when discussing the sensitivities and specificities of the QIAstat-Dx ME panel and the FilmArray ME panel, it would be beneficial to provide the absolute numbers alongside the percentages. Furthermore, it would be helpful to ensure that Table 3 presents these absolute numbers alongside the percentages for both the QIAstat-Dx ME panel and the FilmArray ME panel. This will allow readers to easily compare the performance of the two panels and understand the basis of the calculated percentages.

Results (minor revision): In general, the tables and figures in the manuscript could benefit from additional details to facilitate understanding by the reader. Providing more comprehensive captions, clearer labels, and explanatory notes can significantly improve the interpretability of these visual elements. Ensuring that each table and figure conveys its intended message clearly and completely will enhance the overall quality of the manuscript and the reader's experience.

Discussion (major revision): In line 136, it appears that there may be a broken sentence or missing text, as there seems to be a lack of connection with the subsequent sentences. Please ensure that the entire text of the discussion has been provided and that there are no gaps or discontinuities that might impede the reader's understanding of the arguments presented.

Discussion: The remainder of the discussion appears appropriate to me and effectively addresses the study's context and implications.

Conclusion section (major revision): while the current conclusion is appropriate, it could benefit from including some numerical results that support the findings. For instance, mentioning the specific sensitivity and specificity percentages for the QIAstat-Dx ME panel could provide more concrete evidence for its performance. Additionally, it would be valuable to include conclusions regarding the FilmArray ME panel, highlighting how its performance compares to the QIAstat-Dx ME panel. Emphasizing the importance of conventional methods in conjunction with M-PCR for the comprehensive diagnosis of meningoencephalitis would also strengthen the conclusion. A more detailed conclusion that incorporates these elements would provide a clearer understanding of the study's implications for clinical practice.

I have no further suggestions. Congratulations on your work, and I wish you success with the publication.

Author Response

Reviewer #2.Dear Editor,

I read with great interest the article titled 'Assessment of a new multiplex PCR assays for rapid diagnosis of meningoencephalitis,' written by Genoveva Cuesta and colleagues.

The article describes the analytical performance of a new commercial rapid multiplex PCR system (QIAstat-Dx ME panel) for the detection of infectious agents causing central nervous system infections in clinical samples of cerebrospinal fluid.

In the study, the comparative methods for evaluating the sensitivity and specificity of the QIAstat-Dx ME panel (test method) were clinical conclusions supported by the results of conventional methods, which included cerebrospinal fluid culture, antigen detection, and qPCR for specific viruses. Sanger sequencing of 16S rRNA was used as complementary testing. Samples were also submitted to a second, already established comparative rapid multiplex assay (FilmArray ME panel), which detects the majority of the same pathogens.

Considering the final clinical diagnosis and conventional methods as the gold-standard results, they found that the sensitivities and specificities were 96.5% (CI 95%, 79.8-99.8) and 95.4% (CI 95%, 75.2-99.7), respectively, for the QIAstat-Dx ME panel, and 85.19% (CI 95%, 55.9-90.2) and 57.14% (CI 95%, 29.6-70.3), respectively, for the FilmArray ME panel.

The observed results suggest that the test method not only exhibits good performance in the tested samples but also demonstrates a potential superiority over the comparative rapid multiplex PCR. The good adherence to the final clinical diagnosis also support its higher accuracy.

Given the critical nature of early and accurate identification of the etiological agent causing meningoencephalitis for patient management, documenting the clinical performance of this new assay at different sites is of global importance and great relevance.

Therefore, I recommend the acceptance of this manuscript for publication in the MDPI journal Diagnostics, contingent upon the authors making some adjustments to enhance its scientific suitability for the journal's readership.

These adjustments include ensuring terminology consistency throughout the manuscript, refining phrases for precision and clarity, addressing any potential gaps in the text, including the lack of some results, improving the organization of the text for better flow, explaining some presented results, and enhancing sentence clarity to ensure the message is conveyed effectively. I believe these modifications will significantly improve the manuscript's overall quality and readability.

I will provide the parts of the text that must be revised for these adjustments, following the organization of the text (IMRD), and classify them as major and minor changes. I will also offer suggestions to facilitate a clear and structured revision process, with the understanding that these suggestions are optional for the authors.

Title (suggestion): I suggest moving the word 'rapid' in the title to clarify the nature of the test evaluated in the study, emphasizing its 'bench to bed' approach. For example, 'Assessment of a New Rapid Multiplex PCR Assay for the Diagnosis of Meningoencephalitis. Additionally, if the authors wish to move from a more general title to one that includes the kit name and the specificities of the study, it could be more informative and explanatory.

Thanks. The title has been changed accordingly.

Abstract (minor revision): Line 20 - 'The statistical parameters were calculated.' This statement appears somewhat isolated and lacks detailed information. Could the authors please clarify the specific statistical parameters calculated and how they were applied in the study? If this information is not critical to the understanding of the results, it may be more concise to remove this sentence.

The sentence has been changed. (page 1; lines 24-25)

Abstract (suggestion): I also recommend considering a revision of the abstract to reflect the suggested changes in this review.

Done

Introduction: This section could be expanded slightly, as it currently consists of only one paragraph. Although the study's objective is clearly defined, more details could be included to provide readers with a broader background on the topic. For example, elaborating on the approach to a patient with central nervous system (CNS) infections would help to contextualize the study further. This could include the importance of the biochemical tests presented, as well as the different detection methods used in conventional methods.

Following the reviewer’s suggestion, we have added some information to this section (please see page 1, lines 38-42).

Material and methods (major revision): In Line 53, it is mentioned that 'The remaining samples were frozen at -20°C, then thawed and analyzed by QIAstat-Dx ME panel.' To provide more clarity to the reader regarding storage conditions, it would be helpful to include the range of time the samples were frozen before being tested by the QIAstat-Dx ME panel, if this data is available. This information could clarify storage questions for the reader, even if this storage did not affect the results.

Positive samples have been frozen at -20ºC in a period between 1 and 6 months. We have not detected any difference between the results from QIAstat-Dx ME and conventional methods due the frozen period. For that reason, we consider that any discrepancy with the FA-ME is not caused by the storage conditions. This information has been added into the manuscript ((please see page 2, lines 68-69).

Material and methods (major revision): In Line 54, it is stated that 'CSF cytology and biochemistry results, the final diagnosis of the episode, were collected retrospectively.' It would be beneficial for the reader if the authors could clarify which specific cytology and biochemistry parameters were collected, where they are presented in the study, and why they are important for the study. This additional detail will enhance the understanding of the study's methodology and its relevance to the final diagnosis.

Specific data from all cytological and biochemical CSF characteristics is displayed in the original Table 1.

Material and methods (minor revision): Line 56 - Table 1. Sample processing algorithm. Table 1, titled 'Sample Processing Algorithm,' appears more like a figure than a traditional table. To enhance clarity, I kindly suggest renaming 'Table 1' to 'Figure 1' or an appropriate figure designation.

Done

Material and methods (major revision): In the 'Conventional Methods' and 'Multiplex PCR' sections, the assays' limits of detection are described in different units, which could lead to confusion. For clarification, it would be beneficial to harmonize the units of measurement. For example, in Line 75, the analytical limit of detection for viruses using conventional methods is expressed in copies/mL for herpes viruses, VZV, and enteroviruses, and in IU/mL for HHV-6. In contrast, Line 86 describes the detection limits for viruses in FilmArray ME and QIAstat-Dx ME in TCID50/mL for HSV-1, HSV-2, and enterovirus, and in copies/mL for VZV and HHV-6. It would be helpful to use consistent units across both sections to enhance the reader's understanding of the assay sensitivities.

As we mentioned before this data concerning the limit of detection was provided by the manufacturer and we cannot homogenize it.

Material and methods (minor revision): While the manufacturers of the compared kits are mentioned in the abstract, it would be more appropriate and informative to also cite them in the '2.3. Multiplex PCRs' section. This section details the use of both the FilmArray ME and QIAstat-Dx ME analyses but does not specify the manufacturers' names, which are crucial for replicability and clarity.

Done

Material and methods, and Results (major revision): The '2.4. Statistical analysis' section of the article is well-described and informative, explaining to the reader how all calculations were performed. Although these calculations are traditional and well-known, the detailed description is valuable as it saves the reader from having to consult other references for this information. However, the results of the Cohen's kappa analysis are mentioned only in the abstract and table 4, not in the results section. It would be worthwhile to include the outcomes of the kappa analysis in the text of the results section. This addition would enhance the completeness of the results and provide a clearer understanding of the level of agreement between the different methods used in the study.

The Cohen’s kappa analysis has been now shown in the results section.

Results (major revision): Table 2 – In the 'Results' section, particularly for Table 2, it would be beneficial for the authors to clarify, in the 'Materials and Methods' section, in the results, or in the legend of Table 2, the importance of each analyte measured. The authors should consider whether the biochemical parameters need to be included in this table or could be moved to a supplementary table. Regardless of the authors' decision, details about the importance of these measurements for the final diagnosis can be clarified at the same level of detail observed for the explanation of the statistical analyses. This would clarify their significance for the reader, avoiding the need to consult multiple references to understand their role in the final outcome. Since the final diagnosis is critical for comparison, these details deserve mention.

Following the reviewer’s suggestion, we have added this information as a footnote to the table 1.

Results (major revision): line 117 and table 3 In the second paragraph, when discussing the sensitivities and specificities of the QIAstat-Dx ME panel and the FilmArray ME panel, it would be beneficial to provide the absolute numbers alongside the percentages. Furthermore, it would be helpful to ensure that Table 3 presents these absolute numbers alongside the percentages for both the QIAstat-Dx ME panel and the FilmArray ME panel. This will allow readers to easily compare the performance of the two panels and understand the basis of the calculated percentages.

We think that adding the information request will generate a very cumbersome table. However if the editor thinks it is essential we can modify the table.

Results (minor revision): In general, the tables and figures in the manuscript could benefit from additional details to facilitate understanding by the reader. Providing more comprehensive captions, clearer labels, and explanatory notes can significantly improve the interpretability of these visual elements. Ensuring that each table and figure conveys its intended message clearly and completely will enhance the overall quality of the manuscript and the reader's experience.

We have tried to optimize the tables and figures.

Discussion (major revision): In line 136, it appears that there may be a broken sentence or missing text, as there seems to be a lack of connection with the subsequent sentences. Please ensure that the entire text of the discussion has been provided and that there are no gaps or discontinuities that might impede the reader's understanding of the arguments presented.

Corrected (Page 9, lines 179-180)

Discussion: The remainder of the discussion appears appropriate to me and effectively addresses the study's context and implications.

Thanks

Conclusion section (major revision): while the current conclusion is appropriate, it could benefit from including some numerical results that support the findings. For instance, mentioning the specific sensitivity and specificity percentages for the QIAstat-Dx ME panel could provide more concrete evidence for its performance. Additionally, it would be valuable to include conclusions regarding the FilmArray ME panel, highlighting how its performance compares to the QIAstat-Dx ME panel. Emphasizing the importance of conventional methods in conjunction with M-PCR for the comprehensive diagnosis of meningoencephalitis would also strengthen the conclusion. A more detailed conclusion that incorporates these elements would provide a clearer understanding of the study's implications for clinical practice.

Following the reviewer’s suggestion, we have expanded this section (please see page 9, lines 272-284).

Round 2

Reviewer 1 Report

Comments and Suggestions for Authors

The authors have addressed or discussed their opinions/ given clarification about most of my points raised in the initial report and I think the paper reads better and am happy with the result. The main table is vastly improved. I think the methods are better discussed and the limitations of the paper better desrcibed. 

Comments on the Quality of English Language

I would suggest to double check English grammar and sentence structure as there could still be some improvement - although this is not a major issue.